# A Fusion Model for Saliency Detection Based on Semantic Soft Segmentation

**Jie Tao [1], Yaocai Wu [1], Xiaolong Zhou [2,3,\*], Qike Shao [4,\*] and Sixian Chan [4,5]**

1   Zhejiang Institute of Mechanical & Electrical Engineering, Hangzhou 310000, China;
2   College of Electrical and Information Engineering, Quzhou University, Quzhou 324000, China
3   Key Lab of Spatial Data Mining & Information Sharing of Ministry of Education, Fuzhou 350108, China
4   College of Computer Science and Technology, Zhejiang University of Technology, Hangzhou 310023, China
5   Hangzhou Xsuan Technology Co., Ltd., Hangzhou 310058, China
\*   Correspondence: xionglong@ieee.org (X.Z.); sqk@zjut.edu.cn (Q.S.)

**Abstract:** With the rapid development of neural networks in recent years, saliency detection based on deep learning has made great breakthroughs. Most deep saliency detection algorithms are based on convolutional neural networks, which still have great room for improvement in the edge accuracy of salient objects recognition, which may lead to fuzzy results in practical applications such as image matting. In order to improve the accuracy of detection, a saliency detection model based on semantic soft segmentation is proposed in this paper. Firstly, the semantic segmentation module combines spectral extinction and residual network model to obtain low-level color features and high-level semantic features, which can clearly segment all kinds of objects in the image. Then, the saliency detection module locates the position and contour of the main body of the object, and the edge accurate results are obtained after the processing of the two modules. Finally, compared with the other 11 algorithms on the DUTS-TEST data set, the weighted F-measure value of the proposed algorithm ranked first, which was 5.8% higher than the original saliency detection algorithm, and the accuracy was significantly improved.

**Keywords:** saliency detection; semantic segmentation; foreground extraction; deep learning





## 1. Introduction

People almost always quickly focus their attention on the target area of interest, when they are looking at an image. It is an important mechanism for human visual information, and the target area is called the saliency region. Saliency detection algorithms obtain the saliency region of the image by simulating the visual operation mechanism. With the rapid development of the Internet, the amount of image or video data that needs to be processed is also increasing quickly every day. It may be difficult to complete such a high load of work only by its visual mechanism. Saliency detection can obtain the most saliency target object by removing the meaningless parts of the image. Thus, it can help people to process useful information more quickly and accurately.

Initially, in 1998, Itti et al. [1]. expressed the problem of Visual attention with a computational model and proposed saliency detection algorithms based on the Koch framework. Then, saliency detection was gradually applied to various research fields, such as image retrieval algorithm [2] and object detection algorithm [3], which combined saliency detection. With the rapid development of deep learning, saliency detection has also made a breakthrough. In addition to the application of convolutional neural networks, researchers have also introduced a variety of different algorithms, such as saliency detection based on fully convolutional neural network and low-rank sparse decomposition [4], and saliency detection based on dense weak attention mechanisms [5].

Compared with traditional saliency detection methods using a large amount of prior information, convolutional neural network-based saliency detection algorithm can effectively

improve the ability to identify saliency regions in complex scenes. On this basis, it has incomparable advantages after further integrating other multi-scale features. Wang et al. [6] established a novel saliency detection model using Recurrent Fully Convolutional Neural Networks (RFCNN), which references the recursive framework while retaining the traditional prior knowledge. In addition, the pre-training semantic segmentation strategy is added to improve the training model. Xie et al. [7] proposed a Holistically-Nested Edge Detector (HED) that utilized a fully convolutional neural network (FCN) and deeply supervised nets (DSN) train and predict images, enabling them to learn and extract features at multiple scales and levels. However, saliency detection based on convolutional neural network also has some problems: Convolutional networks extract advanced semantic information by increasing receptive fields through the convolutional layer and pooling layer, but at the same time the accuracy of target boundary is decreased. Furthermore, in saliency detection, the combination of features with different scales is usually used to improve the detection accuracy, but when fusing a variety of features, their contribution to saliency and the difference between high-level features and low-level features are not considered, so the contextual features information cannot be obtained.

Because of such problems, we propose a fusion model for saliency detection based on semantic soft segmentation in this paper. This model can classify Semantic Soft Segmentation (SSS) [8] that is combined with the saliency detection algorithm of Pyramid Feature Attention Network for Saliency detection (PFAN) [9]. SSS [8] is an algorithm used for semantic segmentation, which takes spectral extinction technology to obtain non-local color features, and uses Residual Networks (ResNets) to obtain high-level semantic features; and finally the Laplace matrix is used to classify the pixels in the image to achieve the effect of accurate target segmentation. PFAN [9] designed a Context-aware Pyramid Feature Extraction (CPFE) module to obtain rich contextual features, then combined the Channel-wise Attention (CA) after CPFE feature mapping and Spatial Attention (SA) after low-level feature mapping, and finally used an edge-preserving loss function guidance network to obtain location information.

The main contribution of this paper can be summarized as follows:

(1) Creatively combine the advantages of the two algorithms. The SSS algorithm has good subject edge accuracy in object segmentation, but it only segments the categories in the image and cannot locate the significant region. The PFAN algorithm can efficiently and quickly detect the saliency of the image and obtain the outline and range of the subject, but the accuracy of edge details is not very ideal. The fusion of SSS and PFAN can improve the accuracy of edge segmentation while ensuring the accuracy of the subject.

(2) Detection results of transparent background. The result of the PFAN algorithm is a gray image. The greater confidence (pixel value) means that the probability of the pixel belonging to the subject is higher. However, it is difficult to apply it to the image matting because there are many points with small values in the gray image, which cannot be distinguished only directly by observation. To map to the original image, it needs to be preprocessed. The transparent main diagram module is designed in this paper. Its advantage is that it can intuitively and clearly show the results of object segmentation, to promote the accuracy of the fusion saliency detection algorithm.

(3) Inspired by the fact that PFAN can ensure the accuracy of the rough outline of the subject and SSS can ensure the accuracy of the edge, this paper proposes a saliency detection algorithm based on semantic soft segmentation. Compared with the other 11 advanced algorithms, the weighted F-measure value of our algorithm is better, which can improve the accuracy of saliency detection to a great extent.

The remainder of this paper is organized as follows: Section 2 provides an introduction of the related works; Section 3 describes the detail of our algorithm; the experimental results are presented in Section 4; Section 5 summarizes this paper and discusses further work.

## 2. Related Work

Early saliency detection was mostly analyzed based on low-level features such as manually extracted features and texture colors. Yang et al. [10] proposed a saliency detection method via a graph-based manifold ranking which is carried out in a two-stage scheme to extract background regions and foreground salient objects. Xu et al. [11] proposed a background subtraction algorithm, which used the Gaussian mixture model to extract the texture and color information in the image to comprehensively represent the background. Lin et al. [3] used an adaptive background template and spatial before improving the saliency detection method. Firstly, according to the saliency map of the adaptive background module in the image, the comprehensive saliency map is obtained by judging the difference between the saliency map of the super-pixel block and the background module. Then, according to the aggregation of the salient object, a new spatial prior method was presented. Finally, the final salient map is obtained by fusing the above two saliency maps. Traditional detection methods mostly simulate human intuition for detection. However, it is difficult to obtain high-level semantic information in the image by a priori methods and color features like the above.

With the development of deep learning, various neural network structures have also been widely used in saliency detection, which can better obtain high-level semantic features, that is, it makes up for the shortcomings of traditional detection methods. The Deep Saliency (DS) algorithm proposed by Lee et al. [12], which combines manual features and high-level features, compares the low-level features with the rest of the image. In addition, then, the high-level features of the image are used to judge the saliency of each region of the image by encoding with the VGG network. Zhang et al. [4] combined the full convolution neural network with low-rank sparse decomposition, then transformed the feature matrix and fused the low-rank sparse decomposition method by using the full convolution neural network to obtain the high-level semantic features in the image as a priori information; finally, the saliency map was calculated based on the decomposition results. In addition, the attention mechanism is very suitable for saliency detection because it has a good feature selection ability. For example, Zhang et al. [13] proposed an attention guidance network integrating multi-level information, which can optimize the weakness of the previous backbone network in acquiring semantic features with the mechanism of multi-path circular feedback, then transferred the obtained global semantic features to a lower level. The new two-way messaging model integrated the saliency clues contained in multi-layer features, and the messages transmitted between multiple features predicted the saliency region with transmission rate control. However, the differences between high-level and low-level features may be not noticed when using the attention mechanism in a specific direction. In this paper, the PFAN algorithm uses context-aware pyramid feature extraction (CPFE) and a channel-aware method to obtain context information combined with certain weight allocation [9], which is helpful to generate more accurate detection.

## 3. The Algorithm Presented in This Paper

As illustrated in Figure 1, our saliency detection fusion algorithm based on semantic segmentation is composed of semantic soft segmentation model SSS (shown in the upper block diagram of the flowchart) and a saliency detection pyramid feature attention network PFAN (shown in the lower block diagram of the flowchart). The specific algorithm fusion will be expanded in detail in Section 3.3.

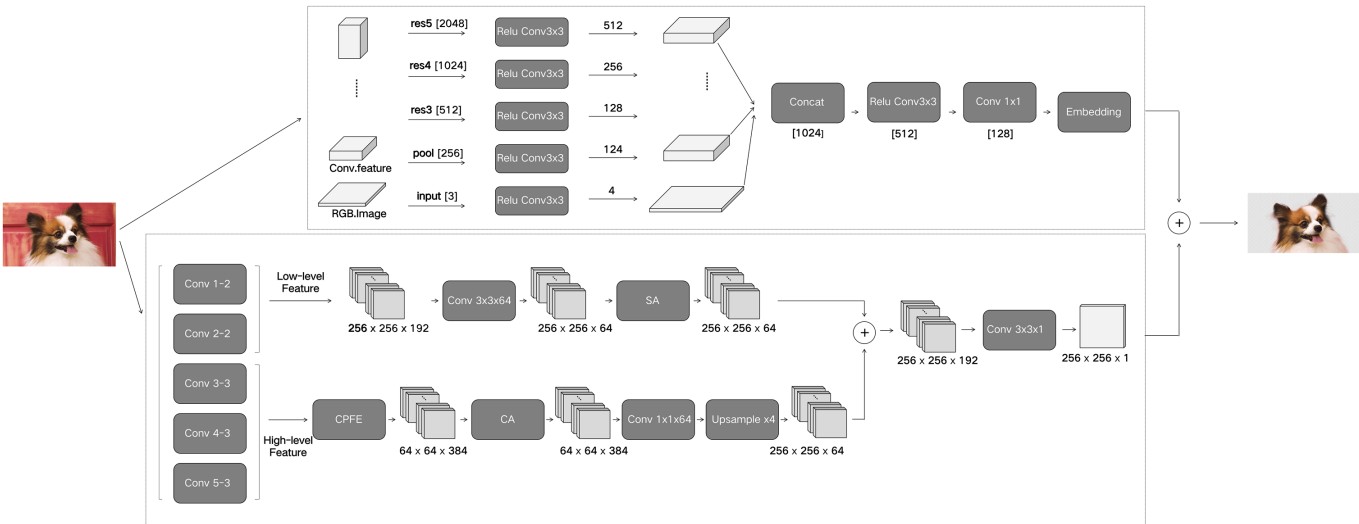

**Figure 1.** Saliency detection algorithm flow based on semantic soft segmentation.

### 3.1. Semantic Soft Segmentation

Semantic soft segmentation is a training algorithm that makes the edge accurate and focuses on the transition region pixels of the main edge. Then, the deep neural network ResNet-101 is used to generate the semantic features of the image, which are presented as 128-dimensional feature vectors. Finally, the soft segmentation is automatically realized by Laplace's matrix decomposition. The flow chart is shown in the SSS section of Figure 1. Firstly, the image information features are collected by two aspects. The texture and color information of the input image is obtained from the the perspective of spectral analysis by spectral extinction; and the other part is high-level semantic information generated by the convolution neural network for scene analysis. Then, the texture, color information, and high-level semantic information of the image are combined to reveal the semantic objects; moreover, the soft transition between them in the feature vector of the corresponding Laplace matrix through a graph structure. Finally, to generate high-quality layers with feature vectors for image editing, a spatially varying layer sparsely model is introduced.

Through the above principle, the semantic soft segmentation greatly improves the accuracy of the subject edge.

#### 3.1.1. Non-Local Color Affinity

In order to represent the relationship between the large ranges of pixel pairs, a guided sampling algorithm based on image transition segmentation is proposed to construct a low-level affine relationship item to represent the correlation characteristics of a large range of pixels based on color. There are two key points in the construction process: one is to generate 2500 super pixels using a Simple Linear Iterative Cluster (SLIC), and the other is to evaluate the affinity of each super-pixel and all super-pixels within 20% of the size radius of the image.

For two super-pixels s and t segmented by distances of less than 20% of the image size, the radiological relation term defining their centroids is defined as:

$$W_{s,t}^c = \frac{erf(a_c - (b_c - \|c_s - c_t\|)) + 1}{2} \tag{1}$$

where $c_s$ and $c_t \in [0,1]$ are the mean color of the super-pixels. $erf$ is a Gaussian error function. $a_c$ and $b_c$ are the rate controlling the affinity drop and the threshold to become 0. (The parameter value used in this paper is: $a_c = 50$, $b_c = 0.05$.) This method essentially ensures the connectivity of color similar areas in complex scene structures.

### 3.1.2. High-Level Semantic Affinity

Non-local color Affinity still represents low-level features, resulting in the frequent merging of image regions with similar colors but belonging to different objects. Therefore, a semantic relationship needs to be added. On the one hand, pixels belonging to the same scene object are encouraged to the group; on the other hand, pixels from different objects are prevented from being grouped. By using a neural network to calculate the corresponding features, the feature vector is generated. DeepLab-ResNet-101 is used as the feature extractor to train the semantic segmentation network on the COCO-stuff dataset. Then, a guided filter is used to align the edge of the feature map generated by the network with the image, and PCA is used to reduce the dimension of the feature map to 3, to adjust fewer parameters and make it easier to construct the image. Finally, the eigenvalue is adjusted to [0, 1] by using normalization processing. After obtaining this eigenvector, this eigenvector is used to define the relationship between the two super-pixels and:

$$W_{s,t}^c = erf\left(a_c - \left(b_c - \left\|\tilde{f}_s - \tilde{f}_t\right\|\right)\right) \tag{2}$$

where $\tilde{f}_s$ and $\tilde{f}_t$ represent the mean eigenvectors of $s$ and $t$, respectively.

### 3.1.3. Creating the Layers

Based on spectral extinction and the convolution neural network, after confirming the input information of two images (nonlocal color relationship and high-level semantic relationship), it is necessary to complete the establishment of the image layer. It mainly consists of two steps: forming Laplace matrix forming and sparse constraints analysis.

**Laplacian matrix forming:** Through the relationship between the two groups of pixels, the Laplace matrix $L$ is constructed by the principle of the least square optimization problem:

$$L = D^{-\frac{1}{2}}\left(D - \left(W_L + \sigma_S W_S \sigma_C W_C\right)\right)D^{-\frac{1}{2}} \tag{3}$$

where $W_L$ is the matrix containing the approximate relationship of all pixel pairs. $W_C$ is the matrix containing the non-local color relationship. $W_S$ is the semantic relationship. $\sigma_S$ and $\sigma_C$ are the parameters controlling the influence of the corresponding matrix (Both of them are set to 0.01). $D$ is a diagonal matrix.

**Sparse constraints analysis:** The feature vectors corresponding to the 100 minimum eigenvalues of the L matrix are extracted, and then the pixels represented by the feature vector f are clustered by K-means. In practice, this method is used to generate 40 layers and remove the unimportant 15–25 layers. Then, the k-means algorithm with $k = 5$ is used to further reduce the unimportant layers represented by the average eigenvector. Such initialization is more consistent with the semantics of the scene and can produce better soft segmentation.

### 3.2. Saliency Detection

Because high-level features contain global context-aware information, they are suitable for correctly locating saliency region, but they can only frame an approximate area. The low-level features contain spatial structure details, which are suitable for locating the boundary, but noise is also introduced. Considering the advantages and disadvantages of high-level features and low-level features, the saliency detection pyramid feature attention network (PFAN) pays attention to both high-level context semantic features and low-dimensional spatial structure features, so as to ensure the extraction of effective features. The algorithm consists of three main steps. The flow chart is shown in the PFAN part of Figure 1. Firstly, the context-aware pyramid feature extraction module (CPFE) is designed to capture rich context semantic features for multi-scale high-level feature mapping; then, the channel attention module (CA) and the spatial attention module (SA) extracted from low-level features are added after the CPFE module; moreover, CA and SA are fused together; finally,

an edge-preserving loss is used to guide the network to learn more details of boundary location information.

### 3.2.1. Context-Aware Pyramid Feature Extraction module (CPFE)

As shown in Figure 2, CPFE takes Conv 3-3, Conv 4-3, and Conv 5-3 of VGG-16 as basic high-level features.

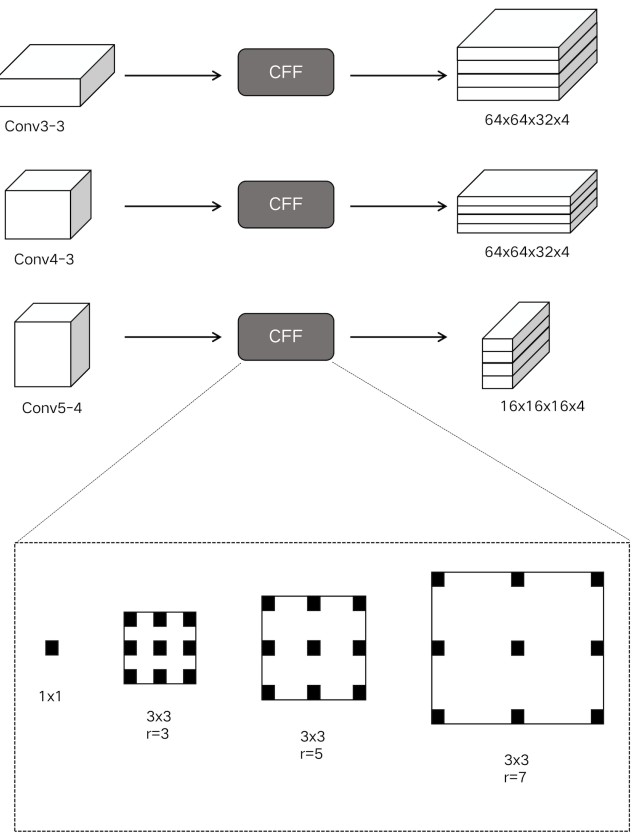

**Figure 2.** Context-aware pyramid feature extraction module framework representation diagram.

Firstly, in order to make the final extracted high-level features meet the scale invariance and shape invariance features, the Atrus convolution with different expansion rates is used to capture the context information, which is set to 3, 5, and 7, respectively. By cascading across channels, the characteristic diagrams from the different Atrus convolution layers are spliced with 1 × 1 dimension reduction features. Then, three different scale features are obtained by using context-aware information, and the two smaller features are sampled upward as the largest one. Finally, all features are cascaded across channels as the output of the context-aware pyramid feature extraction module.

### 3.2.2. Focus Chip Generation

**Channel Attention (CA)**

As illustrated in Figure 3, the appropriate scale and receptive domain are selected by using Channel Attention (CA) to produce saliency regions, which is similar to the SENet method. Advanced features are represented by:

$$f^h \in R_W \times H \times Cas\left(f^h\right) \tag{4}$$

where $R_W$ is the slice width. $H$ is the slice height. $Cas(f^h)$ is the number of channels corresponding to $f^h$.

As illustrated in Figure 3, the Channel Attention method consists of four steps: firstly, the channel eigenvector V is obtained by average pooling of each $f_i^h$; then, two consecutive

full connection (FC) layers are used to fully capture the channel dependency; moreover, to limit the complexity of the model for optimizing generalization, the channel feature vector is encoded by forming two FC layer bottlenecks around the nonlinearity; finally, the coding channel eigenvectors mapped to [0,1] are normalized by using a Sigmoid function.

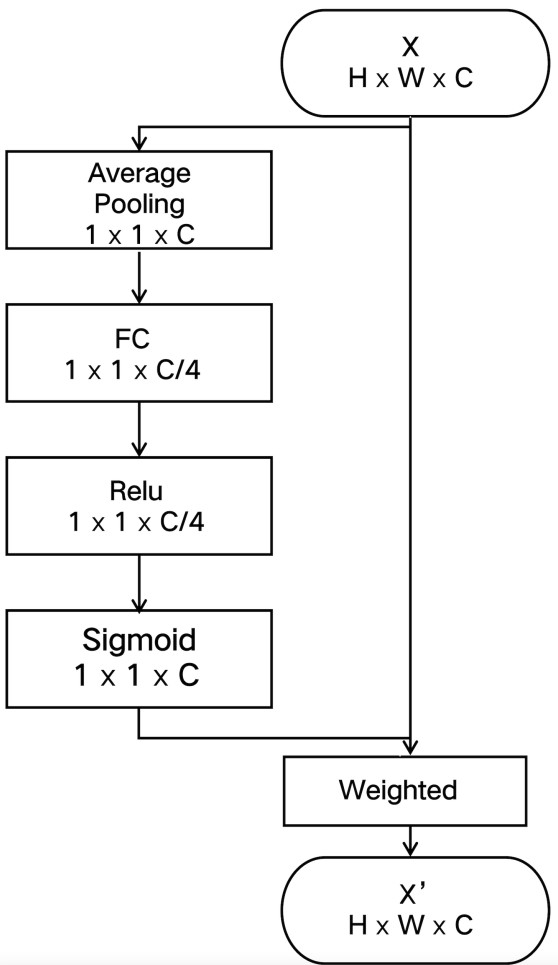

**Figure 3.** The flowchart of Attention mechanism CA.

**Space Attention (SA)**

To generate more precise boundaries for locating saliency regions, low-level features of spatial attention are used to obtain clearer boundary information. Low-level features are expressed as $f^I \in R_w \times H \times C$. The set of spatial positions is represented by $R = \{(x, y) \mid x = 1, \ldots, W; y = 1, \ldots, H\}$. The specific process of the space attention module is shown in Figure 4. To obtain global information without adding parameters, two convolution layers are used, one core is $1 \times k$, and the other core is $K \times 1$, which is used for high-level features to capture spatial concerns. Then, the coding spatial feature map mapped to [0,1] is normalized by Sigmoid operation:

$$C_1 = conv_2\left(conv_1\left(\widetilde{f}^h, W_1^1\right), W_1^2\right) \tag{5}$$

$$C_2 = conv_1\left(conv_2\left(\widetilde{f}^h, W_2^1\right), W_2^2\right) \tag{6}$$

$$SA = F\left(\widetilde{f}^h, W\right) = \sigma_2(C_1, C_2) \tag{7}$$

where $W$ refers to the parameters in the spatial attention block. $\sigma_2$ refers to the Sigmoid coefficient. $conv_1$ and $conv_2$ refer to the convolution layer of $1 \times K \times C$ and $K \times 1 \times 1$,

respectively. $k$ was set to 9 in the experiment. The final output of the block $\widehat{f}^l$ is obtained by weighting with $SA$, which is defined as

$$\widehat{f}^l = SA \cdot f^l \tag{8}$$

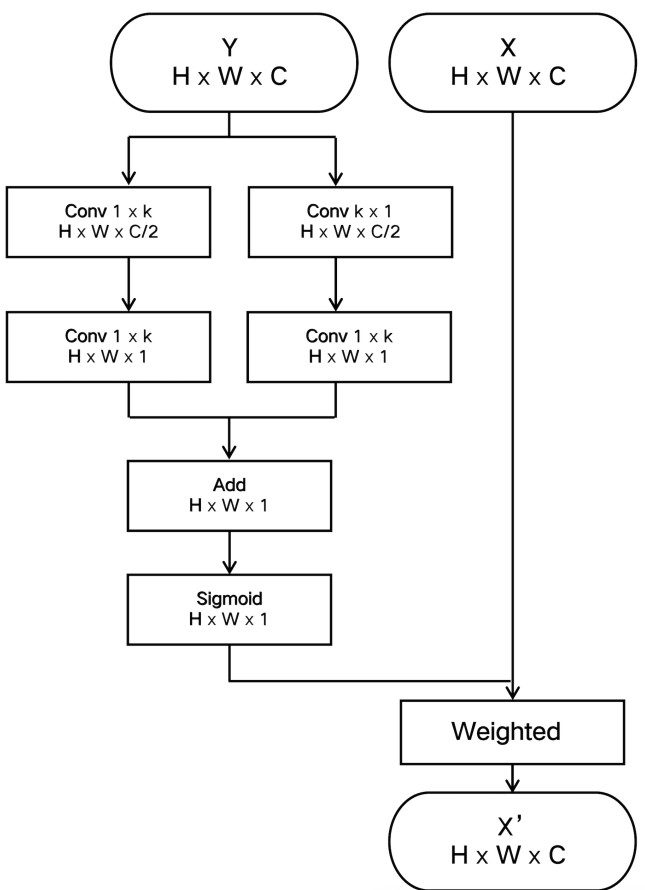

**Figure 4.** The flowchart of Space Attention SA.

### 3.2.3. Loss Function

We utilize the edge preservation loss to guide the network to learn more detailed information in boundary localization. Firstly, the Laplace operator is used to obtain the ground real boundary and saliency map output by the network, and then the cross-entropy loss is used to supervise the generation of saliency object boundary. Laplace operator is a second-difference operator in $n$-dimensional Euclidean space, which is defined as the divergence of the gradient ($f$):

$$\Delta f = \frac{\partial^2 f}{\partial x^2} + \frac{\partial^2 f}{\partial y^2} \tag{9}$$

where $x$ and $y$ are standard Cartesian coordinates of the $xy$ plane. Then, the value is mapped to [0,1] by using absolute operation and the *tanh* activation function:

$$\Delta \bar{f} = \text{abs}\left(tanh\left(conv\left(f, K_{\text{laplace}}\right)\right)\right) \tag{10}$$

Moreover, the cross-entropy loss is used to supervise the generation of significant object boundary formulas:

$$L_B = -\sum_{i=0}^{\text{size}(Y)} \left(\Delta Y_i \log(\Delta P_i) + (1 - \Delta Y_i)\log(1 - \Delta P_i)\right) \tag{11}$$

where $Y$ represents the basic fact graph, and $P$ represents the saliency graph of the network output. Ultimately, the total loss function is their weighted sum of:

$$L = aL_S + (1 - a)L_B \tag{12}$$

where, $a$ represents the balance parameters of positive and negative samples and is set as 0.528 in the training processing.

### 3.3. Saliency Detection Model Based on Semantic Soft Segmentation

The saliency detection algorithm has a good effect on identifying the foreground subject, which can basically recognize the general outline of the foreground in the picture, but the edge of the subject is very fuzzy. The semantic soft segmentation is accurate and clear for the edge details of each class in the image. However, with inaccurate object classification, the foreground subject may be divided into multiple classes or the irrelevant parts are also divided into the foreground subject. Inspired by this, we propose a saliency detection algorithm based on semantic soft segmentation. The model consists of three main steps: first, deal with the results of semantic soft segmentation because, in primitive semantic soft segmentation, each pixel may belong to multiple classes, that is, there are multiple transparencies. Therefore, it is necessary to add loop traversal to the original code to determine that each pixel belongs to only one class; then, the saliency detection result is processed because the original saliency detection result only has the main body contour frame in white, which cannot be combined with soft segmentation. Therefore, the gray image is obtained through the intermediate result of the algorithm operation, and the gray image is binarized to retain the obvious white area of the main body, not just a wheel color contour. Finally, the five matrices are dot multiplied by the gray image of the saliency detection results respectively, and then the required reserved classes and the intersection between the reserved classes are determined. Finally, combining all the parts that keep records is the final required foreground body part.

#### 3.3.1. Processing of Semantic Soft Segmentation Results

In the original code of semantic soft segmentation, each pixel is processed and judged to obtain transparency of each class $a \in [0, 1]$.

Thus, $a = 0$ indicates complete transparency. $a = 1$ indicates complete opacity, and the middle value indicates the degree of opacity. In the final result, multiply the corresponding transparency with different colors to show the classification concretely. The model formula is as follows:

$$(R, G, B)_{\text{input}} = \sum_i a_i(R, G, B)_i \tag{13}$$

$$\sum_i a_i = 1 \tag{14}$$

Since each pixel may belong to multiple classes, that is, there are multiple transparencies, which will interfere with the final result, ideally, it should be determined that each pixel belongs to only one class, which is convenient for the combination of subsequent and saliency detection results.

Five matrices will be generated in the original code, corresponding to five classes respectively. The corresponding value in the matrix is the transparency of the pixel in this class. Therefore, the design algorithm traverses the whole matrix to find the class with the maximum transparency to which each pixel belongs, which is considered to be the class to which it belongs. In addition to this class, the transparency in the other four matrices is set to 0.

### 3.3.2. Processing of Significance Test Results

The final result obtained from the original code of significance detection is a gray image. Only the edge contour of the main body is white, and the rest is black (as shown in the rightmost figure of Figure 5 below).

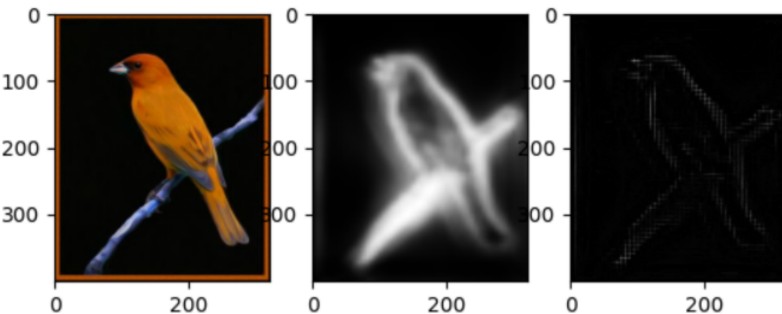

**Figure 5.** Saliency detection contour.

Only the contour information cannot be combined with the results of semantic soft segmentation. Ideally, the main part should be white, not just the outline. Therefore, by analyzing the intermediate results of the operation of the algorithm, we can obtain the desired gray image, and process the gray image at the same time: there may be light white pixels in the black area, which we don't want to keep. Binarize the gray image and keep the obvious white area. Examples of the results before and after processing are shown in Figure 6 (the left side is the execution result of the original significance detection, and the right side is the resulting diagram after binarization of the gray image).

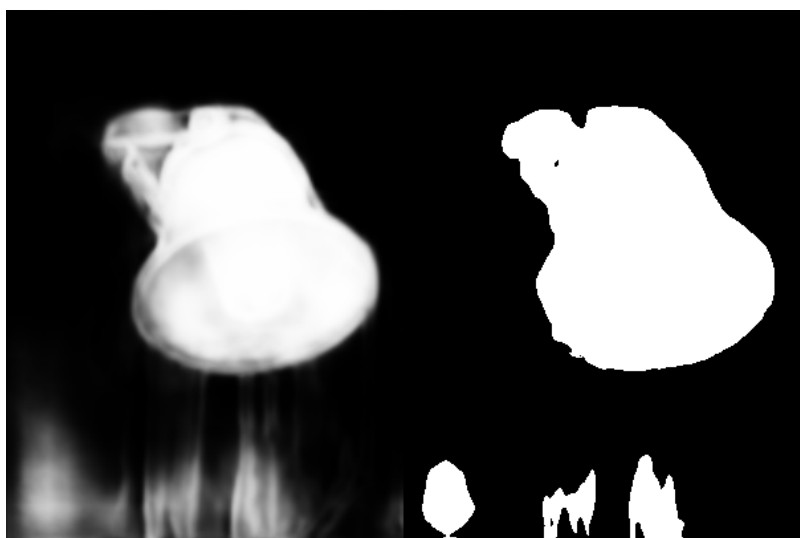

**Figure 6.** Schematic diagram of grayscale result improvement.

### 3.3.3. Algorithm Fusion Processing

The intermediate result of semantic soft segmentation is five two-dimensional matrices, which record the pixel values of five classes, respectively. If the processed result is pixels belonging to this category, there is a value; if the pixel does not belong to this class, it is assigned 0. The specific judgment of whether the pixel belongs to this category is determined by the above transparency.

Then, the five matrices are dot multiplied with the gray image of the sign detection results, respectively. In the result, only the intersection has a value, and the non-intersection is 0. Traverse the matrix multiplied by points to obtain the number of pixels with values, which can be regarded as the area of intersection, marked as $s$. At the same time, the area

size of the class is *small*, and the area size of the foreground body of the gray image is marked as *big*.

Let *SL* be expressed as the ratio of intersection area to class area, which can be obtained by:

$$SL = \frac{s}{small} \tag{15}$$

Let *BL* be expressed as the ratio of intersection area to foreground main area, that is, by the expression:

$$BL = \frac{s}{big} \tag{16}$$

Therefore, if the value of *SL* is large, it is considered that this class belongs to a part of the prospect. In addition, all records of this class are retained; On the contrary, the value of *BL* is judged. If the value of BL exceeds a certain range, the intersection part is retained. The final result is to combine all the parts of the record.

Finally, an example of the saliency detection fusion model result image result based on semantic soft segmentation is shown in Figure 7.

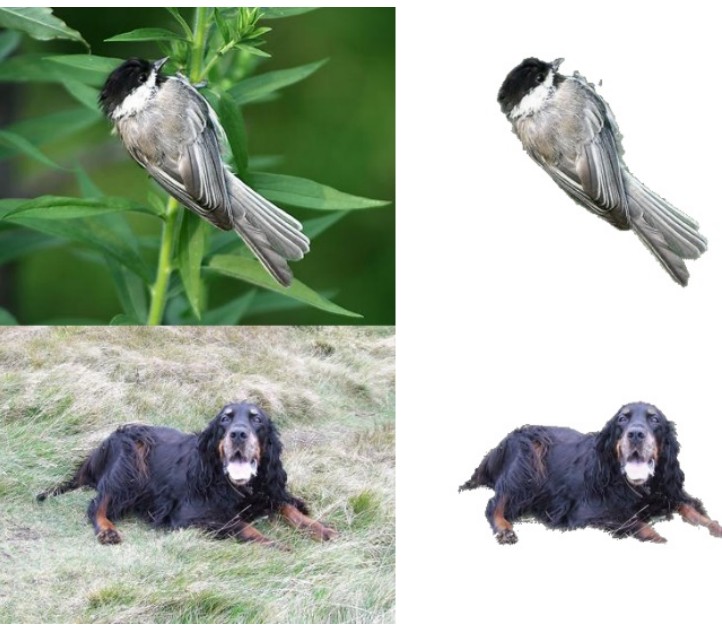

**Figure 7.** Schematic diagram of improved results.

## 4. Experiment

*4.1. Introduction of the Experimental Environment and Data*

### 4.1.1. Training Environment

The training process of the target detection model requires high hardware requirements and needs to use GPU for calculation. Our training environment is shown in Table 1.

**Table 1.** Training environment configuration instructions.

| Product Name | Model | Quantity |
|---|---|---|
| CPU | INTEL I7 8750H | 1 |
| Memory | 16 GB/2666 | 1 |
| graphics card | GTX 1060 | 1 |
| SSD | 512 GB | 1 |

### 4.1.2. Training Dataset and Test Dataset

After repeatedly comparing the existing image segmentation dataset, this paper decides to select the COCO data set and ECSSD dataset for model training and testing.

COCO dataset is large and a rich object detection and segmentation dataset. This data set is mainly intercepted from complex daily scenes, and the target in the image is calibrated by accurate segmentation. Images include 91 types of targets, 328,000 images, and 2,500,000 labels. The number of records is 330k images and 80 object categories; each image has five labels and 250,000 key points.

The ECSSD dataset was established by Shi et al. [14] in 2013 and contains 1000 images, which are obtained from the Internet. The data set is extended by complex scene saliency data set (CSSD). Salient objects contain complex structures and the background has a certain complexity.

There are 11,000 images in the dataset, among which 10,000 images are selected from the COCO dataset and 1000 images from the ECSSD dataset. In this paper, the data set was divided into a training set and a test set in a ratio of 8:2 following academic standards. There were 8800 images in the training set (8000 for COCO and 800 for ECSSD) and 2200 images in the test set (2000 for COCO and 200 for ECSSD). There are no images in both the training set and the test set. The COCO dataset is used to train and test the semantic soft segmentation model, while the ECSSD training set is used to train and test the saliency segmentation model. This paper mainly uses the standard dataset DUTS-Test [15], which includes 10,553 training images and 5019 test images, including very important scenes for saliency detection. In addition, 1000 images were selected to evaluate the detection algorithm.

### 4.1.3. Model Training

In this paper, semantic soft segmentation and saliency detection are integrated, aiming to determine the subject through saliency detection, and then separate the subject through semantic soft segmentation. The latter pays more attention to image edge details processing and finally forms a complete saliency detection algorithm based on semantic soft segmentation. In the process, two models need to be trained. One is to train the model based on the COCO data set, mainly to extract feature vectors; the second is to train the model based on the ECSSD data set, mainly to locate the subject in the image.

The semantic soft segmentation model is designed to extract feature vectors. For the feature vector extraction model, the following methods are adopted: the neural network is used to calculate the corresponding features, to generate feature vectors. Deeplab-resnet-101 is used as the feature extractor, the metric learning method is adopted for network training, and semantic segmentation network training is carried out on the coco-Stuff data set. Then, the guided filter was used to align the edge of the network-generated feature map with the image, and the PCA method was used to reduce the dimension of the feature map to 3, which reduced parameter adjustment and made graph construction easier to handle. Finally, the eigenvector-value is adjusted to [0, 1] through normalization.

The saliency detection model aims to locate the subject in the image. Consider conv3-3, conv4-3, and conv5-3 in VGG-16 as the basic advanced features. In order to make the final extracted high-level features contain the features of scale and shape invariance, atrophic convolution with different expansion rates are adopted, and the expansion rates are set to 3, 5, and 7 to obtain multiple context information. Then, feature maps of different cascaded convolutional layers are combined with $1 \times 1$ dimensionality reduction features cascaded across channels. Subsequently, feature and context-aware information of three different scales are obtained. Finally, they are combined through cross-channel connections as the output of the context-aware pyramid feature extraction module.

### 4.2. Introduction of Evaluation Indicators

Firstly, the evaluation standard of this paper adopts the common accuracy and recall, and binaries the results of the algorithm and the manually marked truth map, selects the gray value of 0–255 as the threshold and uses the following formula to calculate the specific accuracy P and recall R. Here, this algorithm is compared with the PFAN algorithm and is divided into two graphs to compare the accuracy and recall, respectively:

$$P = \frac{|ST \cap GT|}{|ST|} \tag{17}$$

$$R = \frac{|ST \cap GT|}{|GT|} \tag{18}$$

where $ST$ represents the area where the algorithm result is binarized and the value is 1. $GT$ represents the area where the truth graph is binarized and the value is 1.

To comprehensively consider the accuracy rate and recall rate, weighted F-measure ($F_\beta$) is also adopted, and its calculation formula is shown as follows:

$$F_\beta = \frac{(1+\beta) \times P \times R}{\beta^2 \times P + R} \tag{19}$$

where $P$ represents accuracy rate. $R$ denotes the recall rate. $\beta$ is the weight coefficient and set to 0.3, that is, the weight value of accuracy is increased to highlight its influence. In this paper, the accuracy and recall rate of each graph and its corresponding truth graph were calculated first, and then its F-measure value was calculated and averaged to obtain the final result. Finally, the average absolute Error index ($MAE$) is used to show the difference between the result and truth graph by comparison of pixel precision. The calculation formula is as follows:

$$MAE = \frac{1}{M \times N} \sum_{i=1}^{M} \sum_{j=1}^{N} |S(i,j) - GT(i,j)| \tag{20}$$

where $M$ and $N$ represent the length and width of the image, respectively. $S(i,j)$ represents the pixel value in the resulting graph, and $GT(i,j)$ represents the pixel value in the truth graph. By summing the difference between the two and dividing it by all pixel points, the degree to which the resulting graph is close to the truth graph can be obtained, so the smaller $MAE$ means the better results.

*4.3. Results Analysis*

4.3.1. Quantitative Analysis

On the test dataset, the algorithm proposed in this paper is compared with 11 other significant algorithms, including RFCN [6], ELD [12], PAGRN [13], BDMPM [16], GRL [17], Amulet [18], SRM [19], UCF [20], DCL [21], DHS [22] and NLDF [23]. The performance of each algorithm on the DUTS-test data set is counted. Table 2 counts the values of F-measure and Mae corresponding to each algorithm for comparison.

According to the quantitative results, it can be seen that this algorithm is significantly better than other algorithms, ranking the highest score, which verifies the effectiveness of this algorithm. At the same time, it can also be seen in Figure 8 that the results obtained by the algorithm in this paper are higher than the original algorithm, which proves that the integration of SSS greatly optimizes the accuracy of saliency detection. However, the MAE value is not prominent, that is, there are some differences between the resulting diagram and the truth diagram. The reason is that when fusing SSS and PFAN, the intersection part of the two is selected to a certain extent, that is, the more minor part of the saliency region is discarded, to ensure that the final result must be in the significant area of the image because most of these small areas are at the edge of the main body, and the overall accuracy is improved by sacrificing these small areas with uncertainty, This also leads to a further widening gap between the resulting chart and the truth chart, which is the reason why the recall index in Figure 9 does not perform well.

**Table 2.** Comparison of index results in each dataset.

| Methods | DUTS-Test | |
| --- | --- | --- |
| | $\omega F_\beta$ | MAE |
| Ours | 0.9210 | 0.0802 |
| PFAN | 0.8702 | 0.0405 |
| BDMPM | 0.8508 | 0.0484 |
| GRL | 0.8341 | 0.0509 |
| PAGRN | 0.8546 | 0.0549 |
| Amulet | 0.7773 | 0.0841 |
| SRM | 0.8269 | 0.0583 |
| UCF | 0.7723 | 0.1112 |
| DCL | 0.7857 | 0.0812 |
| DHS | 0.8114 | 0.0654 |
| DSS | 0.8135 | 0.0646 |
| ELD | 0.7372 | 0.0924 |
| NLDF | 0.8125 | 0.0648 |
| RFCN | 0.7826 | 0.0893 |

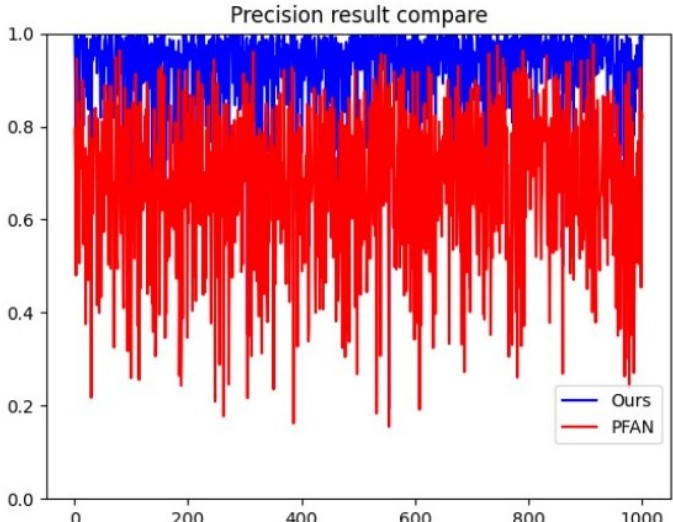

**Figure 8.** The comparison of Accuracy between algorithms.

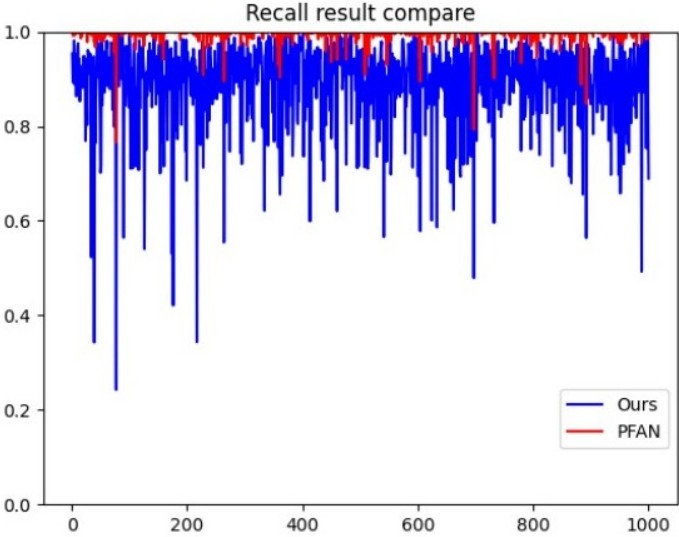

**Figure 9.** The comparison of Recall between algorithms.

4.3.2. Qualitative Analysis

Figure 10 shows the results of the comparison between our algorithm and the other algorithms. In order to make the comparison more intuitive and clear, the resulting diagram of the algorithm in this paper is also processed into a gray image. It can be seen from the comparison diagram that the resulting diagram obtained by other algorithms can ensure the approximate accuracy of the significant area, but the edge details are not very accurate and fuzzy. The edge details of the resulting graph of this algorithm are richer, which significantly improves the accuracy of detection.

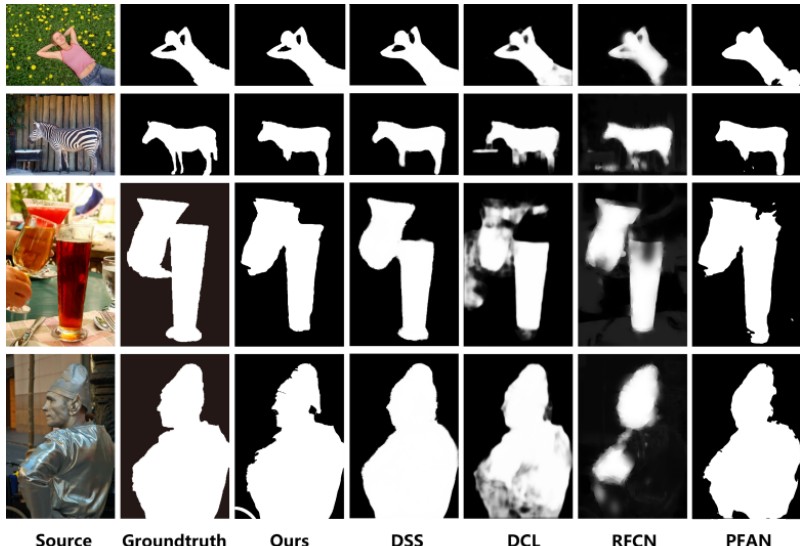

**Figure 10.** Schematic diagram of improved results.

**5. Conclusions**

Aiming at the scene of image subject segmentation, this paper proposed a saliency detection algorithm based on semantic soft segmentation, which effectively combined the advantages of the PFAN algorithm to quickly locate the subject contour range and the SSS algorithm to improve the accuracy of the edge of the segmented subject. This made up for the shortcomings of their respective algorithms. The problems of subject correctness and edge segmentation accuracy of image subject segmentation were solved by our algorithm. Moreover, excellent results on the DUTS-test dataset were achieved. The accuracy of saliency detection was greatly improved. Because of the expansion of the application scope of the proposed algorithm, the main goal of the future work is to optimize the parameters in the fusion,and strive to reduce the MAE value of the result, that is, to narrow the gap with the truth map, and strive to apply the algorithm to practice, such as separating the main body of the video, replacing the photo background, and so on.

**Author Contributions:** Conceptualization, J.T. and X.Z.; investigation, J.T.; methodology, Q.S. and Y.W.; software and validation, S.C.; writing—original draft preparation and funding acquisition, X.Z.; writing—review and editing, J.T., Q.S. All authors have read and agreed to the published version of the manuscript.

**Funding:** This work was supported in part by the National Natural Science Foundation of China (61876168), Joint Funds of the Zhejiang Provincial Natural Science Foundation of China (Grant No. LZJWZ22E090001), and the Key Lab of Spatial Data Mining & Information Sharing of the Ministry of Education (2022LSDMIS02), and the Hangzhou AI major scientific and technological innovation project (2022AIZD0061).

**Institutional Review Board Statement:** Not applicable.

**Informed Consent Statement:** Not applicable.

**Data Availability Statement:** This study did not report any data. We used public data for research.

**Conflicts of Interest:** The authors declare no conflict of interest.

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
