# Peer review of "A Fusion Model for Saliency Detection Based on Semantic Soft Segmentation"

_electronics, doi:10.3390/electronics11172712_

Round 1

Reviewer 1 Report

This is a well-written paper on the interesting topic of saliency detection. The article combines PFAN contour identification with SSS algorithm in a very intelligent way.
The obtained edge segmentation accuracy is pretty good, and
this makes the results a significant contribution to the scientific literature.

Please correct minor spelling errors on

page 3, line 116 ". ,"
        line 121 and 122 ".."

improve grammar for the following sentence in order to make it clear, on page 2, line 54:

 "This model can classify Semantic Soft Segmentation (SSS)[8] is combined with the saliency detection algorithm of Pyramid Feature Attention Network for
Saliency detection (PFAN)[9]."

Author Response

Please find our responses as attached file.

Reviewer 2 Report

This paper proposed a fusion model for saliency detection based on semantic soft segmentation. However, the paper cannot be accepted due to the following reasons.

1. The innovation of this paper is not obvious. Regarding contribution 1, the simple combination of algorithms can not be considered as a contribution.

2. The article spends too much space on basic knowledge, such as the introduction of AM and CNN.

3. There are too many grammatical and typographical errors in the article.

4. The detailed descriptions of variables in the equations are not presented, which makes it difficult for readers to understand the proposed method. 

5. The figures in this paper are too vague for readers to get useful information.

Author Response

(The authors gave the same response as above.)

Reviewer 3 Report

Authors present a saliency detection method by combining existing soft segmentation and pyramid feature attention network for saliency detection in order to improve the performance around object boundary.

Comments:

--The two already existing popular architectures SSS and PFAN are simply combined to add the outputs which is trivial from the novel contribution point of view. Moreover, the exact methods about how do they combine and why enhanced results are expected from this combination is not explained well. If author want to combine the individual benefits of such existing methods, what quantitative and qualitative benefits in terms of accuracy or other metrics are achieved is very important aspect. The analysis in terms of this perspective does not exist in the paper. Moreover, whether that achieved advantage is considerable while loosing the computational complexity and model compactness should be taken into account. This is lacking in the paper. 

---The paper lacks a chronological story in the introduction about the problems with the current saliency detection algorithms and the justifications about why the author actually wanted to do this work, if it wants to solve those problems. 

---The diagram in Figure-1 is not readable. The text and numbers on the arrow could not be understood. Moreover, the loss function block's output from PFAN is added with the embeddings from SSS. What does that mean?

---The English language and writing styles should be updated to correspond the basic requirements of scientific journal. Please avoid the long sentences (Like the first sentence of introduction) and grammar. I suggest a complete re-writing and reformatting of all the text.

Author Response

(The authors gave the same response as above.)

Round 2

Reviewer 2 Report

The revised paper has been improved.